# 1 Impact Study of Increased Radio Occultation Observations during the

# 2 ROMEX Period Using JEDI and the GFS Atmospheric Model

- 3 Hailing Zhang<sup>1,2</sup>, Hui Shao<sup>2</sup>, Benjamin Ruston<sup>2</sup>, John J. Braun<sup>1</sup>
- <sup>1</sup>The Constellation Observing System for Meteorology, Ionosphere and Climate (COSMIC) Program,
- University Corporation for Atmospheric Research, Boulder, CO, USA
- <sup>2</sup>Joint Center for Satellite Data Assimilation (JCSDA), University Corporation for Atmospheric
- Research, Boulder, CO, USA
- Correspondence to: Hailing Zhang (hailingz@ucar.edu)
- **Abstract**. The international collaborative Radio Occultation Modeling EXperiment (ROMEX) 10 project marks the first time using a large volume of real data to assess the impact of increased 11 Global Navigation Satellite System radio occultation (GNSS-RO) observations beyond current 12 operational levels, moving past previous theoretical simulation-based studies. The ROMEX project enabled the use of approximately 35,000 daily RO profiles—nearly triple the number 13 14 typically available to operational centers, which is about 8,000 to 12,000 per day. This study 15 investigates the impact of increased RO profiles on numerical weather prediction (NWP) with 16 the Joint Effort for Data assimilation Integration (JEDI) and the global forecast system (GFS), 17 as part of the ROMEX effort. A series of experiments were conducted assimilating varying amounts of RO data along with a common set of other key observations. The results confirm 18 19 that assimilating additional RO data further improves forecasts across all major meteorological 20 fields, including temperature, humidity, geopotential height, and wind speed, for most of 21 vertical levels. These improvements are significantly evident in verification against both 22 critical observations and the European Centre for Medium-Range Weather Forecasts (ECMWF) 23 analyses, with beneficial impacts lasting up to five days. Conversely, withholding RO data 24 resulted in forecast degradations. The results also suggest that forecast improvements scale 25 approximately logarithmically with the number of assimilated profiles, and no evidence of 26 saturation was observed. Biases in the forecast of temperature and geopotential height over the 27 lower stratosphere are discussed, and they are consistent with findings from other studies in 28 the ROMEX community.

54

55

5657

#### 1 Introduction

31 Global Navigation Satellite System Radio occultation (GNSS-RO) is an active remote 32 sensing technique that measures the refraction of signals transmitted by GNSS and received by 33 instruments aboard low-Earth orbit (LEO) satellites. The pioneering GPS/Meteorology 34 (GPS/MET) mission demonstrated that the GNSS-RO (RO hereafter) technique can effectively 35 probe the Earth's atmosphere, providing profiles with high vertical resolution and accuracy 36 (Kursinski et al., 1997). Since then, the number of RO profiles has increased with the expansion 37 of GNSS beyond GPS (e.g., GLONASS, Galileo, and BeiDou), along with the deployment of 38 more RO receivers aboard new LEO missions, such as the U.S./Taiwan FORMOSAT-39 3/Constellation Observing System for Meteorology, Ionosphere, and Climate (COSMIC-1; 40 launched in 2006) and its successor FORMOSAT-7/COSMIC-2 (COSMIC-2, launched in 41 2019), the European Space Agency (ESA) MetOp series (MetOp-A, 2006; MetOp-B, 2012; 42 MetOp-C, 2018), and the ESA/US Sentinel-6 (launched in 2020). 43 RO data are considered as one of the most impactful observation types in terms of their 44 contribution to the forecast skills in numerical weather prediction (NWP). The positive impact 45 of RO observations on NWP analysis and forecast has been well-documented by numerous 46 NWP centers (Healy and Thépaut, 2006; Bowler 2020; Ruston and Healy 2021; Cucurull 2023; 47 Samrat et al. 2025). Unlike satellite radiance data, RO observations are inherently unaffected 48 by clouds or precipitation and therefore their assimilation in NWP requires no bias correction. 49 Instead, RO observations serve as a reference to anchor the bias correction of satellite radiance 50 data (Healy et al. 2008; Bauer et al. 2014). 51 Since launched in 2019, COSMIC-2 has been steadily providing approximately 6,000 RO 52 profiles daily, primarily between 45°S and 45°N. Other government missions in polar orbits 53

profiles daily, primarily between 45°S and 45°N. Other government missions in polar orbits contribute around 2,000–4,000 daily profiles globally to NWP centers, with the number varying depending on the data ingested by each center. More recently, the emergence of commercial RO providers, such as GeoOptics Inc. (Pasadena, CA, USA), PlanetiQ (Golden, CO, USA) and Spire Global Inc. (Boulder, CO, USA), Yunyao Aerospace (China), and Aerospace Tianmu (China) have further expanded RO data availability. These commercial sources supplement

operational capabilities with additional profiles, depending on purchase agreements. With this expanded data volume, NWP centers have recently been enabled to explore the impact of assimilating slightly more than 10,000 RO profiles per day. Several NWP centers have demonstrated that the relative impact of RO data in NWP has grown alongside the increasing availability of profiles. Bowler (2020) at the Met Office assessed the RO data produced by Spire and stated that the forecast impact of increasing the RO data volume is roughly proportional to the logarithm of the total amount of GNSS RO data assimilated. Ruston and Healy (2021) reported a novel finding that COSMIC-2 data improve the tropical tropospheric humidity forecasts in both the Navy Global Environmental Model (NAVGEM) and the European Cent—re for Medium-Range Weather Forecasts (ECMWF) Integrated Forecasting System (IFS). The assimilation of COSMIC-2 and Spire observations was found beneficially in both the ECMWF and Met Office's NWP systems (Lonitz et al. 2021). Cucurrul (2023) demonstrated that COSMIC-2 observations have a significant impact in the forecast improvement of temperature and winds in the tropics.

While these studies demonstrated the valuable impact of increased RO profiles in operational NWP systems, the potential benefits of even larger data volumes were only explored through theoretical simulations. Harnisch et al. (2013) used an ensemble of data assimilations (EDA) approach to evaluate the change of RO data impact as a function of observation numbers. They demonstrated that saturation was not found with 128,000 simulated RO profiles per day. With a global observing system simulation experiment (OSSE) study, Privé et al. (2022) found that the assimilation of 100,000 daily RO profiles did not reach the impact saturation in the hybrid four-dimensional ensemble variational data assimilation system and Global Earth Observing System (GEOS) model.

Clearly, the number of RO observations currently utilized in real-time NWP operations remains significantly below the potential demonstrated in these simulated studies. Meanwhile, a large portion of RO observations from commercial providers is not purchased and remains unassimilated in operational systems, highlighting that the full impact of RO from both government and commercial providers has yet to be fully realized. Since May 2022, the International RO Working Group (IROWG; https://irowg.org), one of the scientific advisory working groups of the Coordination Group for Meteorological Satellites (CGMS), has led an

international collaborative effort, the Radio Occultation Modeling Experiment (ROMEX; Anthes et al. 2024), to explore the impact of RO observations by collecting as many profiles as available from both commercial and government providers during the testing period. Specifically, ROMEX has collected nearly 35,000 daily profiles during the experimental period (September to November 2022), whereas there are about 12,000 daily profiles available in the real-time NWP operations at the U.S. National Oceanic and Atmospheric Administration (NOAA). ROMEX provides a unique opportunity for both NWP centers and the research community to evaluate impacts of increased RO numbers using large quantities of real RO observations for the first time.

The overarching objective of this study is to demonstrate forecast impact through the assimilation of the increased RO data. We aim to quantitatively assess these data impacts with respect to operational implementations, while leveraging advanced features for enhanced performance. Specifically, this study utilizes the Joint Effort for Data assimilation Integration (JEDI; Trémolet and Auligné 2020) for data assimilation and the NOAA Global Forecast System (GFS) for forecasting. Given JEDI is the next generation data assimilation system for operations at NOAA, the National Aeronautics and Space Administration (NASA), the U.S. Naval Research Laboratory (NRL), and other NWP centers worldwide, this ROMEX study offers additional benefits by demonstrating JEDI's capabilities and providing insights for ongoing transitions to operations.

This manuscript is organized as follows: Section 2 summarizes ROMEX and the RO observations used for this study; Section 3 introduces the GFS forecast model, the JEDI data assimilation system, and the experimental design; Section 4 presents the evaluation of the ROMEX RO data impact using the JEDI-GFS system; Section 5 presents a summary of the work.

#### 2 ROMEX and GNSS RO observations

ROMEX is an IROWG initiative designed to evaluate the impact of increasing radio occultation (RO) data volume using real observations from both government and commercial missions, extending beyond current operational capabilities. ROMEX involves approximately

30 international agencies and research institutions, including data providers, processing centers, NWP centers, universities, and research institutes. A complete list of ROMEX participants is available on the ROMEX website (<a href="https://irowg.org/ro-modeling-experiment-romex/">https://irowg.org/ro-modeling-experiment-romex/</a>). The outcomes of ROMEX provide guidance to CGMS for formulating recommendations to space agencies on RO mission planning and coordination. Additionally, ROMEX results offer valuable insights for RO data providers, processing centers, and NWP centers to enhance data retrieval techniques and improve the assimilation of RO data in weather forecasting.

Through dedicated data agreements with commercial RO providers, the ROMEX effort was able to access data not covered by existing global licenses held by NOAA, NASA, or EUMETSAT from their respective commercial purchases. The ROMEX dataset includes commercial RO data otherwise unavailable to the public, along with publicly available data from sources such as the UCAR COSMIC Data Acquisition and Access Center (CDAAC), NOAA, NASA, and EUMETSAT. Due to the involvement of multiple processing centers and data providers, different processing versions of the data were available to support validation and processing studies.

Our objective is to evaluate the impact of an increased volume of RO profiles on analyses and forecasts, rather than comparing the performance or characteristics of various missions. Early data evaluation already shows the quality of these data is relatively comparable for NWP applications (Marquardt 2024; Anthes et al. 2025). The available profiles are categorized into two groups based on their sources: base missions (hereafter, BASE), and the supplementary missions. The base missions include COSMIC-2, Metop-B/C, Kompsat-5, PAZ, Sentinel-6, TerraSAR-X, and TanDEM-X, all of which are commonly available in near-real-time and assimilated for operational NWP systems participating in the ROMEX project. This selection helps avoid discrepancies arising from differences in NWP operational configurations. The supplementary missions consist of GeoOptics, PlanetiQ (Kursinski 2025), Spire (Nguyen 2025), Yunyao (Cheng 2025), Tianmu (Tang 2025) and Fengyun (Liao 2024). On average, approximately 35,000 daily profiles (8,000 from base missions and 27,000 from supplementary missions) were available during the ROMEX period (hereafter, ROMEX). All these data are distributed through the EUMETSAT Radio Occultation Meteorology Satellite Application

Facility (ROM SAF; Marquardt 2024). This study uses version 1.1 of the dataset. Particularly,

COSMIC-2, TanDEM-X, TerraSAR-X, Kompsat-5, and PAZ are processed by UCAR, and

Metop-B/C, Spire, Yunyao, PlanetIQ, and Sentinel-6 are Processed by ROM SAF from excess

phase. Fengyun and Tianmu are encoded to BUFR format by ROM SAF from the "atmPrf"

files provided by the National Space Science Center (NSSC), China.

To further quantify the impact of the increased profile volume, the EUMETSAT ROM SAF provided a sub-dataset referred to as ROMEX20K, in which the average daily number of profiles is 20,000, meaning that ROMEX20K has approximately 12,000 supplementary profiles per day in addition to the BASE dataset.

Figure 1 presents the total number of RO profiles in each 5° × 5° latitude-longitude grid for September 2022, the testing period of this study. Specifically, Fig. 1a–c shows the number of BASE profiles, supplementary mission profiles, and all available ROMEX profiles, respectively. Fig. 1d-e displays the supplementary profiles used in the ROMEX20K and all ROMEX20K profiles. Fig. 1f-g shows the ratio of supplementary profiles used in ROMEX20K with all supplementary profiles, and the ROMEX20K profiles relative to the total number of ROMEX profiles, respectively.

The total number of BASE profiles (Fig. 1a) peaks in the tropics and decreases poleward, primarily due to the dominance of COSMIC-2. The supplementary profiles (Fig. 1b), however, are more evenly distributed across the mid-to-high latitudes. Overall, the combination of all available profiles (Fig. 1c) results in a relatively uniform global distribution geographically. The supplementary profiles used in the ROMEX20K sub-dataset kept a higher portion over the northern hemisphere and southern mid-to-high latitudes than over the tropical regions (Fig. 1f). Combined with the base profiles, ROMEX20K has a better coverage over the tropics than other regions of the globe and the fewest profiles over the southern polar regions (Fig. 1g).

**Figure 1:** Total number of RO profiles in September 2022, re-gridded to a 5° × 5° latitude–longitude grid, for the following data sets: (a) base missions (BASE), (b) all supplementary missions, (c) all ROMEX missions (ROMEX, equivalent to a+b), (d) supplementary missions in the ROMEX20K, (e) all missions used in the ROMEX20K configuration (ROMEX20K, equivalent to a+d). Panels (f) and (g) show the ratio of total profile counts between (d) supplementary in ROMEX20K and (b) total supplementary missions, and between (e) ROMEX20K and (c) all ROMEX missions, respectively.

#### 3 Forecast model, data assimilation, and experimental design

#### 3.1 Forecast model

The Global Forecast System (GFS) is NOAA's medium-range operational global weather prediction model, developed and maintained by the National Centers for Environmental Prediction (NCEP). It is part of the Unified Forecast System (UFS), a community-based, coupled Earth modeling framework designed to integrate research and operational weather modeling for more consistent and advanced forecasts (Zhou et al. 2022). This study used the atmospheric forecast model component of UFS, and not the entire suite. Further, the next

planned release version 17 implementation<sup>1</sup> (GFSv17; GFS hereafter) was employed in this study. This latest version of GFS is continuing to use the Finite-Volume Cubed-Sphere (FV3; Lin 2004) dynamical core, and also incorporates significant upgrades in parameterizations for atmospheric processes such as cloud microphysics (Stefanova et al. 2022; Meixner et al. 2023), in comparison to the current operational implementation at NCEP. The global forecasts for this study are configured at a horizontal resolution of approximately 25 km with 127 vertical levels extending up to 80 km (C384L128). This is half of the operational resolution and is standard practice for pre-implementation testing at NCEP and by associated researchers.

#### 3.2 Data assimilation

This study uses the Joint Effort for Data assimilation Integration (JEDI; Trémolet and Auligné 2020) to fulfill the data assimilation component. Led by the Joint Center for Satellite Data Assimilation (JCSDA), JEDI was initiated in 2017. As the project has grown partners now include NOAA, NASA, NRL, the U.S. Air Force, the NSF National Center for Atmospheric Research (NCAR), UK Met Office and developers from universities. JEDI has been interfaced to various models, including the GFS, through the JEDI-FV3 component (<a href="https://github.com/JCSDA/fv3-jedi/">https://github.com/JCSDA/fv3-jedi/</a>), allowing the partner agencies using FV3 core-based systems to implement JEDI in real-time applications.

The observation operators for JEDI are developed within an abstracted and generic coding layer known as the Unified Forward Operator (UFO). A generic design makes UFO model-agnostic and allows it to be used in a play-and-plug manner through configuration files. Currently, UFO includes six GNSS RO operators, four for bending angle and two for refractivity, contributed by different partners to replicate the implementation in their respective NWP systems. The four bending angle operators include one based on the operational NCEP Bending Angle Model (NBAM; Cucurull et al. 2013), the Met Office's bending angle operator (Burrows 2014, Burrows et al. 2014), and both one-dimensional (ROPP1D; Healy and Thepaut

<sup>&</sup>lt;sup>1</sup>GFSv17 has not been implemented in the operation as we started this work to the best of our knowledge. We checked out the branch prototype/hr3 in August 2024 from thttps://github.com/ufs-community/ufs-weather-model

2006) and two-dimensional (ROPP2D; Healy et al. 2007; Healy 2014) operators interfaced via **ROM-SAF** Occultation the Radio Processing Package (ROPP; https://romsaf.eumetsat.int/ropp), which are used operationally by NRL and ECMWF, respectively. Additionally, two refractivity operators are included, following implementations from the Met Office and NCEP (Cucurull et al. 2007, Buontempo et al. 2008). Most NWP centers use 1D bending angle operators operationally, considering both its impacts and computational efficiency. While a detailed comparison of these operators is performed in a separate effort, we use the ROPP1D operator with the default JEDI configuration that was based on the current implementations by partner agencies. UFO also includes associated quality control (QC) procedures and observation error models, allowing creation of a consistent treatment to those used in operational applications at other centers.

#### 3.2.1 GNSS RO forward operator

Assuming the atmosphere is horizontally homogeneous and spherically symmetric, ROPP1D computes the bending angle,  $\alpha$ , by vertically integrating the refractive index from the model background, as shown in Equation 1 (Healy and Thepaut 2006).

$$\alpha(a) = -2a \int_a^\infty \frac{1}{\sqrt{x^2 - a^2}} \frac{d \ln(n)}{dx} dx \tag{1}$$

a is the observed impact parameter, n is the modelled refractive index, and x = nr is the product of the refractive index and the radius value r of a point on the ray path. Note that the impact parameter is a geometric quantity representing the closest distance between the straight-line trajectory of a GNSS signal and the Earth's center; it is the actual coordinate in RO assimilation. Impact height, defined as the difference between the impact parameter and the local radius of curvature of the Earth, is referred to as the vertical coordinate when presenting RO space results.

The model background information is extracted for each observation point along the RO profile, valid at the horizontal location of its corresponding tangent point. Therefore, the vertical drift of tangent points is fully accounted for. However, ROPP1D does not consider the integrated effect of atmospheric bending along the ray path, as is done in ROPP 2D. The comparison between these two operators will be a separate work.

#### 3.2.2 RO observation error and quality control

Observation error and QC procedure are two crucial parameters in DA. The observation error accounts for measurement uncertainty, representativeness error, and forward operator error (Bormann 2015). Accurate modeling of observation errors is essential for appropriately weighting RO observations relative to other data types and the background error. Meanwhile, QC procedures are closely linked to both the forward operator and observation uncertainty, as they aim to remove observations with large departures that may result from forward operator limitations or various sources of measurement error. As such, observation error characterization, QC, and the forward operator are tightly interconnected in the assimilation of RO observations.

The first QC procedure applied in this study checks the quality flag provided by the data providers; observations labeled "non-nominal" are excluded. The second procedure, a background check QC, rejects observations if the difference between the simulated and observed values exceeds three times the specified observation error.

We applied the observation error model used in the NRL designed system (Ruston and Healy 2021) that is run operationally at Fleet Numerical Meteorology and Oceanography Center (FNMOC). Figure 2 shows the observation errors (in percentage) averaged over a random day, as functions of latitude and impact height. In this scheme, observation errors are defined as a percentage of the observed values and decrease linearly with increasing impact height, reaching a minimum of 1.25 % at the "minimum error height". A damping factor is applied to account for latitudinal variation. In JEDI implementation, the error is specified as 20 % at the surface (impact height is 0) at 0° latitude and is reduced away from the equator following the cosine of latitude. The minimum error height also varies with latitude, decreasing from 12 km at the equator to 5.333 km at the poles – again modulated by the cosine of latitude. Above the minimum error height, the observation error is specified as the greater of 1.25 % of the observation value or 3 microradians. Thus, the fixed 3 microradians floor can correspond to a relative error much larger than 1.25 %.

**Figure 2:** Percentage observation errors (%) for all RO observations on September 1, 2022, using the NRL error model.

## 3.3 Experimental design

Four sets of experiments, namely, noRO, BASE, ROMEX20K, and ROMEX, were conducted using the JEDI-GFS system over a one-month period in September 2022, which represents one of the three months of the ROMEX period (September–November 2022). All experiments assimilated a common set of observations from the Global Data Assimilation System (GDAS) archive, including conventional data from radiosondes, aircraft, and surface stations, as well as atmospheric motion vectors, scatterometer wind vectors and satellite radiances from AMSU-A and ATMS measurements aboard multiple satellites available during the study period.

These four experiments differ only in the volume of RO profiles assimilated. The noRO experiment excludes RO data entirely. The BASE experiment assimilates only publicly available RO profiles, totaling approximately 8,000 per day. ROMEX20K and ROMEX assimilate approximately 20,000 and 35,000 daily RO profiles, respectively, based on the corresponding datasets. All RO profiles are assimilated from the surface up to 55 km using the same configuration, i.e., the same observation error specification and QC. Differences in forecast skill among these experiments illustrate the impact of enhanced RO data volume available during the ROMEX period.

All experiments were initialized at 0000 UTC on September 1, 2022, using a 6-hour forecast as background from the operational GFS system at NCEP. The JEDI-GFS system was then cycled every 6 hours, with a 6-hour assimilation window and background fields provided by the previous forecast cycle. Data assimilation was performed using JEDI's hybrid three-dimensional variational (3DVar) method, with 40 ensemble members taken from the NCEP global ensemble forecast system. The data assimilation minimization was performed on a so-called dual-resolution grid: the background and forecasts used the C384 grid, while the minimization was carried out on the C192 grid.

#### 4 Results and evaluation

This section compares the results of all experiments to assess the impact of RO observations on forecast skill. Short-to-medium range forecasts are evaluated against observations and model analyses. In observation space, common evaluation metrics include observation—minus—background (OMB) and observation—minus—analysis (OMA) statistics, whose mean values are often referred to as background bias or analysis bias, respectively. In model space, forecast skill is assessed by comparing model forecasts and ECMWF analyses (FMA) at analysis grid points. Three basic metrics, root-mean-square error (RMSE), standard deviation (STDV), and mean bias, are calculated for OMB/OMA or FMA over the entire experimental period.

To further evaluate the impact of each experiment relative to the reference (noRO or BASE), two additional metrics are adopted to illustrate the impact of an experiment relative to the reference experiment. The first is the normalized difference of a given metric between the experiment and the chosen reference (Eq. 2), where a negative value indicates improvement and a positive value indicates degradation. The second is the mean absolute error reduction (MABR; Eq. 3), which compares the absolute biases between experiments. A negative MAB R reflects a beneficial bias reduction relative to the reference experiment, while a positive value indicates a detrimental increase.

Normalized difference = 
$$100\% \times \frac{M(Exp.) - M(Reference)}{M(Reference)}$$
 (2)

MABR = 
$$100\% \times \frac{|\text{Bias}(\text{Exp.})| - |\text{Bias}(\text{Reference})|}{|\text{Bias}(\text{Reference})|}$$
 (3)

### 4.1 Evaluation in observation space

Statistics in RO observation space are first calculated to evaluate the performance of the JEDI-GFS system in assimilating the large volume of real RO data from ROMEX. Because RO bending angle observation values span a few orders of magnitudes vertically, the OMB statistics are presented in a normalized format, i.e., OMB/B (B is a 6-h forecast from the previous cycle). Figure 3 shows the RMSE and bias from the ROMEX experiment. RMSE results (Fig. 3a) show that the assimilation produces a reasonable agreement between the bending angle observations and both the 6-h forecast and analysis, with lower RMSE in the analysis than in the background. Biases are also notably reduced after assimilation especially between 3 and 12 km impact heights, when comparing OMA to OMB (Fig. 3b), demonstrating the assimilation's effectiveness in correcting background errors in this key region.

**Figure 3:** (a) RMSE and (b) Bias of OMB (RO bending angle observation minus 6-h model forecast) normalized by the backgrounds (OMB/B; solid) and OMA (RO bending angle observation minus analysis) similarly normalized (OMA/B; dashed), of the ROMEX experiment in September 2022.

Fig. 4 shows the normalized difference in STDV for the fit of temperature, specific humidity, and wind speed forecasts to radiosonde observations. All three experiments show smaller STDV than noRO across all variables and levels. The STDV reduction relative to noRO increases with height, reaching a maximum near 200 hPa for temperature and wind, and at around 700 hPa for humidity. Comparing all three RO experiments (ROMEX, ROMEX20K and BASE), the forecast improvements, or STDV reduction, increase with the growing volume of RO data. ROMEX generates the largest reduction among the three RO experiments. For example, the STDV reduction of temperature forecast at 200 hPa are 3.2%, 5.3%, and 6.8% for BASE, ROMEX20K, and ROMEX respectively. However, the difference between ROMEX and ROMEX20K is negligible in the lower troposphere (below 800 hPa) for temperature (Fig. 4a), and near the surface for wind speed (Fig. 4c). Note that RO data provide only information on the mass fields, wind forecasts are not directly impacted by the assimilation of RO data. Rather, they are impacted through the background error covariance between state variables in DA and the dynamic adjustment through the month-long cycles. Despite this indirect influence, the positive impact of RO data on wind forecasts seen here is very significant.

All RO experiments reduced noRO bias in temperature forecasts between 700 hPa and 100 hPa (Fig. 5). Experiments with additional RO data assimilation tend to cool nearly the entire troposphere, up to 0.17 K (ROMEX vs. NoRO), as indicated by the RO experiment curves lying to the right side of the NoRO curve. In contrast, it warms the air above the jet-stream layer around 200 hPa, where the RO experiment curves shift to the left of the NoRO curve (Fig. 5a). However, the RO experiments produce larger humidity biases compared to noRO. Assimilating more RO data results in a drier atmosphere, while withholding RO data leads to a wetter one (Fig. 5b). The dry effect of ROMEX reaches a maximum of approximately 0.1 g kg<sup>-1</sup> at 700 hPa, relative to NoRO.

**Figure 4:** Normalized difference in STDV (%) of the RO experiments relative to noRO for the 6 h model forecasts verified against radiosonde observations of (a) temperature, (b) specific humidity, and (c) wind speed. Overlaid bars are the standard deviations of the normalized STDV difference.

**Figure 5:** Bias of all experiments for the 6 h model forecasts verified against radiosonde observations (OMB) of (a) temperature (unit: K) and (b) specific humidity (unit: g kg<sup>-1</sup>).

The impact of assimilating ROMEX RO data on other critical observing systems is also examined to understand whether the assimilation of RO data can indirectly enhance the assimilation of other observation types. Figure 6 presents the normalized difference in STDV, relative to noRO, for the fit to aircraft temperature and wind speed observations. Consistent forecast improvements are observed in both fields, aligning with the verification result from radiosonde data. Figure 7 presents the percentage difference in the number of assimilated observations relative to the BASE experiment for six observation types (radiosonde, aircraft, surface, scatterometer, satellite winds or atmospheric motion vectors, and GNSS-RO

observations), by comparing ROMEX20K, ROMEX, and noRO. BASE serves as the reference experiment to facilitate the inclusion of RO data in the comparison. Note also that the three RO experiments assimilate different RO datasets, which does not favor a direct comparison of the total number of assimilated RO observations across experiments. Therefore only the RO observations from the BASE dataset assimilated in all experiments are considered for the bars labeled "GNSSRO" in Fig. 7. The observations passing quality control in the OMB statistics are counted through the entire month for each type. Both ROMEX20K and ROMEX assimilate more data than BASE across all data types used in this study, whereas noRO assimilates clearly fewer. For example, ROMEX20K and ROMEX assimilate 0.88% and 0.59% more radiosonde observations than BASE, respectively, while noRO assimilates 0.73% less. This indicates that assimilating additional RO data brings the model analysis and short-range forecast closer to these observations and enables the use of more observational data. This approach clearly shows that both ROMEX and ROMEX20K increased the assimilation of BASE RO data compared to the BASE experiment.

**Figure 6:** Fractional STDV difference (%) of experiments relative to noRO for the 6 h model forecasts verified against aircraft observations (OMB) of (a) temperature and (b) wind speed. Overlaid bars are the standard deviations of the fractional STDV difference.

**Figure 7:** Fractional observation count difference (%) of experiments relative to BASE for key observing types (radiosonde, aircraft, surface, scatterometer, satellite winds or atmospheric motion vectors, and RO observations from base missions) assimilated in all experiments. Note experiment BASE is used for reference here to account for the RO observation. Only the base RO mission observations are counted for the RO calculation.

**Figure 8:** STDV reductions with the number of daily RO profiles for (a) temperature and (b) wind speed at 250-, 400-, 500-, and 700-hPa, and (c) specific humidity at 400-, 500-, and 700-hPa. X-axis is the experiments, and y-axis the fractional STDV difference relative to noRO. For illustration purposes, positive numbers are reductions for this figure.

To consolidate the STDV reductions for the forecast of temperature, specific humidity and wind at various levels, Figure 8 is presented to show the 6-h forecast STDV reductions for the various experiments that exhibit progressive increases in daily RO profiles at key pressure levels, following the NWP verification exchange guidance provided by ROMEX. Using noRO as the benchmark, the STDV reduction increases approximately logarithmically with the growing number of profiles, consistent with the findings of Lonitz (2025). Overall, no consistent evidence of saturation was found. However, the extent of this non-saturation could be influenced by the specific DA and forecast systems.

## 4.2 Evaluation in model space

This sub-section evaluates the impact of ROMEX data on short-to-medium range forecasts against the ECMWF analysis over the one-month experimental period. The first assessment focuses on the noRO experiment by comparing its results with that of BASE, which represents the consequences of losing or withholding all RO data. Figure 9 presents the zonal-mean of the STDV and bias of the 6-h temperature forecasts of BASE as a function of pressure levels. Also shown are the differences in STDV and bias between noRO and BASE, with statistical significance at the 95% confidence level. The BASE experiment exhibits large temperature STDVs in the tropical tropopause and lower stratosphere (above 50 hPa), as well as in the lower troposphere, especially in the Southern Hemisphere high latitudes (Fig. 9a). When RO data are withheld (noRO), significant forecast degradations (increasing STDV errors relative to the ECMWF analysis) occur above 850 hPa, with a maximum increase in STDV, about 0.4 K, centered near 50 hPa over the tropics. BASE's temperature bias exhibits multiple patterns, including prominent negative values in the tropical tropopause, positive values above and below the tropopause in the low latitudes, and negative values in the lower tropospheric at high latitudes (Fig. 9b). noRO amplifies BASE's existing biases: negative values become more negative, and positive values become more positive, particularly in the mid-to-upper tropical troposphere and near the mid-latitude tropopause. The negative impact of excluding GNSS RO observation as shown in noRO is also observed in the verification of other key parameters such as humidity and wind speed (not shown), therefore the following sections will focus on the presentation of the impact of the two ROMEX datasets relative to BASE.

437438

**Figure 9:** (a) Zonal-mean STDV (shaded), and (b) bias (shaded) of the 6-h temperature forecasts (unit: K) of BASE verified against ECMWF analysis. Overlaid contours are the differences between noRO and BASE (noRO–BASE; unit: K; interval: 0.04 K) in (a) STDV and (b) bias at 95% significance level. Solid/dashed curves represent positive/negative values respectively in both panels. In panel (a), positive values for contours indicate noRO forecast degradation relative to BASE, while negative values indicate improvement. In panel (b), opposite signs between the contours and shading indicate improvement in noRO relative to BASE, while matching signs indicate degradation.

Figure 10 compares the two ROMEX experiments, ROMEX and ROMEX20K, with respect to BASE in terms of STDV (Figs. 10a, c, and e) and MAB R (Figs. 10b, d, and f) for 6-h temperature forecasts, and also includes a direct comparison between ROMEX and ROMEX20K. As introduced earlier, negative MAB R values indicate reductions in absolute bias relative to BASE (i.e., improvement), while positive values indicate increased bias (i.e., degradation). Hashed areas indicate regions where the results are statistically significant at the 95% confidence level. Overall, both ROMEX and ROMEX20K exhibit significant STDV reductions relative to BASE between 850 and 50 hPa. Substantial forecast improvements are observed in the Southern Hemisphere's mid-to-upper troposphere, the tropical tropopause region, and the middle troposphere and stratosphere of the Northern Hemisphere's high latitudes. One exception is that ROMEX produces increased STDVs above 50 hPa over the Southern Hemisphere middle latitudes (Figs. 10a and c), which is linked to the warming effect introduced by the additional ROMEX RO data as seen in the verification against radiosonde temperatures (Fig. 5a). The non-significant degradations over the southern hemisphere at around 950 hPa are likely caused by the terrain height mismatch between the forecasts and the ECMWF analyses. ROMEX outperforms ROMEX20K except for the regions of above 50 hPa over the Southern Hemisphere middle latitudes (Fig. 10e). This is consistent with the detrimental effect of increasing RO data as discussed earlier. Meanwhile, ROMEX and ROMEX20K demonstrate beneficial effects of assimilating additional RO data by showing overall reduced MABR values below 50 hPa, particularly in the tropical tropopause and the middle troposphere at low latitudes. On the other hand, both experiments exacerbate biases in the stratosphere above 50 hPa (Figs. 10b and 10d), which is again attributed to the warming effect introduced by the assimilation of ROMEX observations. Overall, assimilating additional ROMEX RO data improves short-range temperature forecasts in terms of both STDV and MABR when verified against ECMWF analyses, with the exception of the lower stratosphere.

**Figure 10:** Differences in STDV (a) and (c) and MABR (b) and (d) of the two ROMEX experiments relative to BASE for the 6-h temperature forecasts (unit: K) verified against ECMWF analysis, and difference in STDV (e) and MABR (f) of ROMEX relative to ROMEX20K. Hashed areas overlaid

indicate regions of 95% statistical significance level. Positive/negative values of the MABR differences indicate the experiment is farther/closer to the ECMWF analysis.

Figure 11 presents the STDV and bias of the 6-h specific humidity forecasts of the BASE experiment, verified against the ECMWF analysis (Figs. 11a–b), along with the differences in STDV and MABR between both the ROMEX and ROMEX20K experiments and BASE (Figs. 11c–f), and between ROMEX and ROMEX20K (Figs. 11g–h). The STDVs of BASE are largest in the tropical 850 hPa level and decrease gradually with height and toward the poles (Fig. 11a). Both ROMEX experiments show reduced STDV extending from the surface to 300 hPa (Figs. 11c and e), indicating improved forecast performance. Notably, ROMEX yields additional STDV reductions compared to ROMEX20K in the low-to-mid troposphere over the tropics (Fig. 11g). In terms of bias, BASE exhibits a three-layer structure in the tropics, with positive bias near 700 hPa, negative bias around 900 hPa, and another positive bias near the surface (Fig. 11b). Both ROMEX experiments mitigate the positive bias above 700 hPa (Figs. 11d and f), while ROMEX achieves further bias reduction in the 700–400 hPa layer, particularly over tropical regions (Fig. 11h). Overall, both ROMEX and ROMEX20K outperform BASE in terms of humidity forecast skill, with improvements in both STDV and bias, and ROMEX demonstrating an additional advantage over ROMEX20K.

**Figure 11:** (a) STDV and (b) bias of BASE 6-h specific humidity forecasts (unit: g kg<sup>-1</sup>) verified against ECMWF analysis, and differences in STDV (c) and (e) and MABR (d) and (f) of the two ROMEX experiments relative to BASE, and difference in STDV (g) and MABR (h) of ROMEX relative to ROMEX20K. Hashed areas overlaid indicate regions of 95% statistical significance level. Positive/negative values of such differences indicate the experiment is farther/closer to the ECMWF analysis.

**Figure 12:** (a) STDV of BASE 6-h wind speed forecasts (unit: m s<sup>-1</sup>) verified against ECMWF analysis. Differences in STDV for the two ROMEX experiments relative to BASE (b) and (c); and (d) ROMEX relative to ROMEX20K. Hashed areas overlaid indicate regions of 95% statistical significance level. Positive/negative values of such differences indicate the experiment is farther/closer to the ECMWF analysis.

The same diagnostics were also applied to wind speed. Figure 12 displays the 6-h wind speed forecast results: the STDV from BASE (Fig. 12a), the STDV differences between each ROMEX experiment and BASE (Figs. 12b-c), and between the two ROMEX experiments (Fig. 12d). The areas of the largest improvement of wind speed are primarily over the Northern Hemisphere lower stratosphere, the tropical tropopause, and the Southern Hemisphere middle troposphere. Systematic biases are not observed in the wind field and are therefore not presented.

The impact of ROMEX RO assimilation on medium-range forecasts of the JEDI-GFS system is further assessed. Five-day (120-h) forecasts, initiated at each 00Z cycle during September 2022, are examined at 24-h intervals for both experiments. Similar to the short-range evaluation, STDV is calculated against the ECMWF analysis and forecasts of various lead times. The STDV difference with the ECMWF verification is calculated as functions of forecast lead time. Evaluations are conducted in three regions, i.e., the Northern Hemisphere

(NHX, 20°N–80°N), the Tropics (TRO, 20°S–20°N), and the Southern Hemisphere (SHX, 20°S–80°S).

Figures 13-15 illustrate the differences in STDV between ROMEX20K and BASE and between ROMEX and ROMEX20K, as a function of forecast lead time, for temperature, specific humidity, wind speed and geopotential height, respectively. ROMEX20K exhibits reduced STDV, or improved forecast skill, across all three regions throughout the atmosphere above the surface for the temperature 0–5 day forecasts. These reductions persist through the 5-day forecast period, while decaying with lead time (Figs. 13a-c). The most beneficial impacts in the TRO and NHX regions are around 150 hPa, with larger than 8 % improvement at the initial forecast time, and are less than 1 % at day 5 (Figs. 13a-b). In the SHX region, larger than 10 % improvement is observed in a broad layer between 200 and 500 hPa at the initial time, which decays to 2-3% at day 5 (Fig. 13c). With additional data assimilated, ROMEX leads to further improvement in temperature forecast over the lower-to-upper troposphere, lasting up to 3 to 5 days (Figs. 13d-f). The detrimental impacts across the three regions are primarily limited to 20 or 50 hPa with relatively small positive STDV differences toward longer forecast hours for TRO and NHX, and slighter larger values above 50 hPa for SHX than the other two regions. It also shows that the near surface forecast in SHX does not gain benefit from the additional RO assimilation, and the benefit gained above the surface only sustains in the first 3–4 days (Fig. 13f).

ROMEX's degradation relative to ROMEX20K in the upper levels of SHX extends slightly downward with time. This is closely linked to a warming bias in the lower stratosphere over the mid-latitudes of the Southern Hemisphere, which is introduced by the assimilation of additional RO data (also shown in Figs. 10d, 10f, and 5a).

**Figure 13:** Difference in STDV for temperature forecast of (a-c) ROMEX20K relative to BASE, and (d-f) ROMEX relative to ROMEX20K, verified against ECMWF analysis as a function of forecast lead time for region of (a) and (d) NHX (20°N–80°N, (b) and (e) TRO (20°S–20°N), and (c) and (f) SHX (20°S–80°S).

Similar to the temperature forecasts, the positive impact of assimilating additional ROMEX data on specific humidity forecasts is sustained through five days, with the greatest improvement at the initial time that diminishes rapidly with lead time (Figs. 14a–c). For example, forecast improvements around 500 hPa at the 0-h lead time exceed 6% in all three regions, but decrease to less than 1% at 96-h in NHX and at 72-h in TRO. In SHX, forecast improvements are maintained throughout the troposphere over the 5-day period, with approximately a 2 % reduction in STDV at 120-h. The humidity forecast skill of the increased RO assimilation aligns with that of Prive et al. (2022), in which they stated that the dominant baroclinic process in the SHX winter may account for its longer time scale for improved forecast. As more RO data are assimilated, ROMEX's positive impact on top of ROMEX20K extends to 5 days in NHX and TRO, but only 2 days in SHX. The relative degradation starts at the surface and propagates upward from day 0 onward (Fig. 14f).

Figure 14: Same as Fig. 13, but for specific humidity forecast.

Figure 15: Same as Fig. 13, but for wind speed forecast.

The impacts of assimilating increased RO data on wind forecast are overall positive from ROMEX20K results for the three regions (Figs. 15a-c). Larger forecast improvements, such as more than 5% STDV reductions in the first three days for NHX, are seen in the lower stratosphere for all regions. Forecast degradations are only noticed after 96-h below 50 hPa (Fig. 15a). TRO are mostly positive as well, except slightly increasing negative impacts around 50 hPa beyond 48 h and forward (Fig. 15b). SHX shows all positive impacts from ROMEX20K. Unlike NHX and TRO with the largest positive impacts at upper levels, SHX wind improvements from ROMEX20K are presented for all levels, including 4% improvement between 600–1000hPa for the first 24 h. Also noted is that the impacts of RO data assimilation do not diminish as rapidly as was observed in the temperature and humidity forecasts. For example, the maximum improvement in the NHX troposphere is reached at the 48-h lead time (Fig. 15a).

# 5. Summary and discussion

- This study investigates the impact of increased RO profiles as part of the international collaborative ROMEX project. The current RO profiles available to operational centers are about 8,000–12,000 daily depending on the volume purchased from commercial providers. Earlier studies demonstrated that saturation was not reached with even 128,000 daily profiles. For the first time, the ROMEX project enabled the use of approximately 35,000 daily RO profiles to explore this further with Observing System Experiments (OSEs).
- As part of the ROMEX NWP efforts, this study contributes to building consensus on the impact of increased volumes of RO observations and to addressing the risks associated with potential loss of RO capabilities across NWP centers, specifically within the GFS framework NOAA's operational forecasting system. At the same time, the study leverages advanced features of JEDI to enhance performance, serving as a valuable platform to evaluate JEDI as the next-generation data assimilation system.
- Four sets of experiments were conducted over a one-month period in September 2022: noRO, BASE, ROMEX20K, and ROMEX. All experiments assimilated a common set of conventional observations, cloud-motion vectors, and satellite radiances, differing only in the amount of RO data assimilated. The BASE experiment assimilated only the publicly available

RO profiles (~8,000 per day), while noRO excluded RO entirely. ROMEX20K and ROMEX assimilated approximately 20,000 and 35,000 daily RO profiles, respectively. The actual number of RO profiles per day varies depending on quality control procedures.

The results show that assimilating additional RO profiles significantly improves forecast skill for all key meteorological fields, including temperature, humidity, geopotential height, and wind speed, for most of vertical levels. Forecast improvements were evident in verification against both the critical observations and ECMWF analyses, with impacts lasting up to 5 days (maximum forecast range in the experiments). For example, the STDV reduction of temperature 6h forecasts at 200 hPa, relative to noRO, was 5.3% for ROMEX20K and 6.8% for ROMEX when verified against radiosonde observations. Conversely, withholding RO data led to forecast degradations, with a maximum STDV increase of approximately 0.4 K near 50 hPa over the tropics. The results also suggest that forecast improvements scale approximately logarithmically with the number of assimilated profiles, and no evidence of saturation. These results were achieved without any additional tuning of the data assimilation system. All quality control procedures and observation error specifications for RO data used the default, generic configurations implemented in JEDI for testing purposes. The positive outcomes therefore underscore the consistency and robustness of the RO data quality, and demonstrate that assimilating a large volume of RO observations is both feasible and beneficial.

However, this effort also revealed areas requiring further investigation. In particular, the assimilation of additional RO increases biases in temperature within the lower stratosphere. The ROMEX20K and ROMEX experiments introduced a cooling effect throughout much of the troposphere and a warming effect above 200 hPa, leading to increased forecast biases relative to the ECMWF analysis. Such warming/cooling effect leads to substantial biases in geopotential height in the BASE experiment, as shown in Figure 16a, plotting as a function of forecast lead time over the Northern Hemisphere (with similar results for the Southern Hemisphere and the Tropics), verified against the ECMWF analysis. It shows that the BASE is positively biased to the ECMWF forecast above 350 hPa to 500 hPa during the first 120 h and negatively biased below. Figures 16b-c illustrate the differences in geopotential height MAB between ROMEX20K and BASE, and between ROMEX and ROMEX20K. The assimilation of additional RO data reduces the biases of BASE below 50 hPa (predominantly

negative MAB) up to 5 days. However, above 50 hPa, the assimilation of additional RO data introduces degradation, reflected as positive MAB.

626627

Figure 16: (a) Bias of geopotential height forecast (shaded; unit: geopotential meter or gpm) for BASE, verified against ECMWF analysis as a function of forecast lead time for region of NHX (20°N–80°N), and the mean absolute bias (MAB) difference between (b) ROMEX20K and BASE, and (c) ROMEX and BASE. A negative MAB (blue) reflects a beneficial bias reduction relative to BASE, while a positive value (red) indicates a detrimental increase.

The sources of these biases are still under investigation. Geopotential height forecast degradation has also been observed by other NWP centers, including the Met Office (Bowler and Lewis 2025), ECMWF (Lonitz 2024; 2025), and the Environment and climate change Canada (ECCC; Aparicio 2025). ECMWF shows a 2 to 8 m decrease in the 72-h geopotential height forecast, with the assimilation of ROMEX data, in the troposphere and stratosphere (Lonitz 2025). The Met office also shows up to 2.5 m negative bias in the 500 hPa geopotential height forecast due to the extra ROMEX observations (Bowler and Lewis 2025). It is worth noting that the bias presented in this study is verified against the ECMWF, and the BASE is already biased negatively in the lower-to-middle atmosphere and positively aloft (Fig. 16a). Adding additional RO data in our system leads to an overall bias degradation in the lower stratosphere. This degradation occurs at higher altitudes than in the ECMWF and Met Office results. The ECMWF analyses used as a reference were produced with the regular volume of RO data assimilated and therefore may not represent the best possible results achievable with the full ROMEX dataset. The ECMWF analyses themselves may contain inherent biases, some of which are model-related. Further, the biases can also arise from data processing procedures (e.g. Anthes et al. 2025), and assimilating large volumes of data may amplify such impacts.

Note this is the first instance in which the volume of assimilated RO data has nearly tripled, and the interactions between RO data and other observations are not yet fully understood. The Met Office (Bowler and Lewis 2025) and NRL (Christophersen and Ruston 2025) have applied bias correction to RO observations in their systems, which appears to enhance the impact of ROMEX on the forecast, particularly for geopotential height. Though the source of bias are not fully clear, it is still possible to account for them through QC and/or observation error estimation (which includes the forward operator errors) to mitigate their impacts. A separate study is under way to further investigate the bias sources and mitigate the issues shown above, with particular focus on adaptive QC procedures in the upper troposphere and lower stratosphere. It should also be mentioned that the one-month experimental period is relatively short due to the limited computing resources, whereas the ROMEX project recommended experiments performed throughout the three-month ROMEX testing period. In addition, the system did not include hyperspectral infrared sounders or geostationary radiances although a few satellite observations, including AMSU-A, ATMS, atmospheric motion vectors, were assimilated. While we believe the study demonstrates the benefit of increased RO observations using the current JEDI and GFS atmospheric forecast model, results may differ in a fully operational configuration.

To conclude, the assimilation of ROMEX RO data has an overall significantly positive impact in the JEDI-based system. Although no saturation was observed even with the full ROMEX data, the 20K subset significantly improves forecast skill, consistent with recommendations from IROWG-10 (Shao et al. 2025) and the second ROMEX workshop. The combination of the ROPP1D forward operator, the NRL observation error model, and generic quality control, within the JEDI framework, not only enabled the successful assimilation of increased data volume in this study, but also lays the groundwork for future exploration and optimization.

*Author contribution*. HZ and HS co-designed the experiments, built the end-to-end system, and jointly prepared the manuscript. HZ conducted the experiments and verifications and created the figures. HS and BR prepared the datasets and provided guidance throughout the

- project. All authors contributed to the interpretation of the results and the review of the
- manuscript.
- *Competing interests.* None of the authors has any competing interests.
- Acknowledgments. This work was sponsored by the NSF grant 2054356, NASA grant
- C22K0658, and the NOAA Science Collaboration Program Agreement B under grant number
- NA23OAR4310383B. We thank Drs. Rick Anthes at UCAR and Xuanli Li for their
- suggestions and discussions which helped improve the work. Authors would like to
- acknowledge EUMETSAT/the Radio Occultation Meteorology Satellite Application Facility
- (ROM SAF) to facilitate ROMEX data sharing and processing. We would like to acknowledge
- high-performance computing support from the Derecho system (doi:10.5065/qx9a-pg09)
- provided by the NSF NCAR, sponsored by the National Science Foundation.
- Code and Data Availability Statement. The ROMEX RO profiles are provided through
- EUMETSAT ROM SAF. Other types of observations are publicly available at the NSF
- NCAR's Research Data Archive (RDA; https://rda.ucar.edu/datasets/), which contains a subset
- of the NCEP Global Data Assimilation System observations. The JEDI code of the presented
- work is available at https://github.com/JCSDA.

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
