# Peer review of "Impact Study of Increased Radio Occultation Observations during the"

_EGUsphere, 2025_

## Author Comment (AC1)

This paper gives a very nice overview of the impact of assimilating additional GNSS-RO data and shows logically and systematically the improvements ROMEX data has on the forecast scores using the JEDI-GFS system. First, the impact on short range forecasts are shown by comparing short range forecasts to observations. Secondly, the impact of short and medium range forecast scores are shown with using the IFS as a reference. It is impressive to see that 20 K and full ROMEX show systematically an improvement to e.g. humidity and wind in the short-range; which is consistent to what observations are seeing. Also, for longer lead times the impact is quite substantial by the additional assimilation of GNSS-RO data. However, also some detrimental impacts are illustrated and well documented in this paper. In general, the paper reads nicely and is well structured. I recommend to accept this publications with some minor revisions; stated below:

Thank you very much for your valuable feedback. We greatly appreciate your positive comments on our work. In response to the general and specific comments, we have carefully reviewed and revised the manuscript accordingly. A point-by-point response is provided below in blue for clarity.

**General comments**

General question: "Which version of processing was used for Yunyao or other RO data in your study?" Maybe mention that somewhere.

We have added the processing description in Section 2 with a reference to Marquardt (2024), as shown below.

"All these data are distributed through the EUMETSAT Radio Occultation Meteorology Satellite Application Facility (ROM SAF; Marquardt 2024). This study uses version 1.1 of the dataset."

**Specific Comments**

p2, l53: I understand you mention only the commercial data operationally assimilated but maybe also mention the Chinese companies (also used in the ROMEX studies). I am not sure if they are assimilated by CMA nowadays?!

Thank you for the comment. Chinese companies, Yunyao and Tianmu, have been added.

P7, Fig.1: I was wondering what is shown here. The total number of RO profiles over the month of September for every 5x5 box? Or the number of average daily profiles over that month?

It is the total number of RO profiles over the month of September 2022 for each 5x5 box. We have revised the figure caption.

Also, I think it would be better to use a radially symmetric kernel to estimate the number density for the following reason. If one compares the number for 5x5 lat/lon boxes the area covered over the Tropics is much bigger (and more chances to have RO data) than over the Poles. This would give a wrong impression of where the most data is located.

We agree that the areas covered by 5×5° lat/lon boxes vary significantly across latitudes. To avoid the misleading impression, we have replotted Figure 1 using an equal-area projection. This approach allows us to visualize the variation in box areas while still preserving the total number of profiles over the one-month period. Figure 1 has also been refined with a different color bar and an updated caption, incorporating the suggestions from the other two reviewers.

P9, l221-223: This is true but one has to admit that the horizontal location of that observation point can be different to the tangent point horizontal location- hence, we have ROPP 2D. Maybe mention that.

If we understand correctly, the reviewer is pointing out that a RO observation assigned to the tangent point actually reflects the integrated effect of atmospheric refraction along the entire ray path. This is correct, and we appreciate the opportunity to clarify. In the revision, we have noted the limitations of the 1D approach and mentioned the ROPP 2D method as shown below.

"However, ROPP1D does not consider the integrated effect of atmospheric bending along the ray path, as is done in ROPP 2D. The comparison between these two operators will be a separate work."

P11, l252: Mention that September 2022 this is not the full ROMEX period

Thank you for your comment. It is mentioned in the revision.

P13, l323: Maybe indicate the magnitude of this cooling/drying for ROMEX.

The cooling/drying effects are mentioned now as shown below.

"Experiments with additional RO data assimilation tend to cool nearly the entire troposphere, up to 0.17 K (ROMEX vs. NoRO), as indicated by the RO experiment curves lying to the right side of the NoRO curve."

"The dry effect of ROMEX reaches a maximum of approximately 0.1 g kg$^{-1}$ at 700 hPa, relative to NoRO."

**Technical comments**

Throughout the manuscript correct the spelling of "centre" in ECMWF.

Thank you for catching it! All are corrected in the revised manuscript.

p1, l12: add "daily" after "35,000" and before "RO profiles"

Thank you for pointing this out. We have corrected all occurrences.

p2, l30: Replace "RO" with "GNSS-RO". RO is just the way it is measured - it doesn't fit the remainder of this sentence, as it describes GNSS-RO.

p2, l42: Again I would use GNSS-RO to be really accurate but of course you could mention in the text that with RO data you mean GNSS-RO data. (also l.53)

Thank you for the comment. In the revised version, we use "GNSS-RO" at its first appearance and introduce 'RO' as a shorthand thereafter.

P4, l94: Change "improvement" to "impact" or "change"

We have revised it.

 P5, l141: add "daily" after "35,000" and before "profiles". This keeps coming up at more occasions throughout the manuscript when 35000 profiles are mentioned. Please check.

All occasions are checked and revised.

---

## Author Comment (AC2)

Summary

This manuscript evaluates the impact of RO observations from ROMEX on the NWP. Using the GFS model and JEDI assimilation system, results show that assimilating additional RO data can further improve the global weather forecasts, with beneficial impacts lasting up to five days. The result also indicates that there is no saturation for the forecast improvement. The manuscript is well written, and the experimental design and results are solid. The manuscript could be a valuable contribution to the communities of RO, observations, assimilation, models, and forecasts. I have several comments below.

Thank you very much for your valuable comments. We greatly appreciate your positive comments on our work. We have reviewed and revised the manuscript accordingly in response to your comments and questions. A point-by-point response is provided below in blue for clarity.

1. Section 2 and Figure 1, it is unclear about the differences among ROMEX, ROMEX20K, ROMEX sub dataset. Based on the words, I would guess Fig. 1a + Fig. 1b = Fig. 1c, and Fig. 1a + Fig. 1d = Fig. 1e, while Fig. 1d is not included in Fig. 1b. But based on Fig. 1, it is hard to imagine Fig. 1a + Fig. 1d = Fig. 1e. It would be nice to clearly describe the dataset, which could be consistent with the assimilation experiments.

   We have revised the figure caption and panel titles for clarity. The figures have also been replotted with different color bars, incorporating the comments from the other two reviewers.

2. 'Quality control' is defined as 'QC' at line 211, so that QC can be consistently used later on (e.g., line 229, 236, 351...).

   Thank you so much for catching it. Revisions have been made.

   How to QC the RO observations are discussed in the manuscript. Does it use the 3 times of standard deviations of the observation error, or something else?

   Yes, we use the 3 times of standard deviations of the observation error. A paragraph has been added to describe specific QC procedures, as shown below.

   "The first QC procedure applied in this study checks the quality flag provided by the data providers; observations labeled "non-nominal" are excluded. The second procedure, a background check QC, rejects observations if the difference between the simulated and observed values exceeds three times the specified observation error."

3. I have a curious question about the bias introduced by assimilating the RO. Since the observation error at high altitudes is already large (Fig. 2), which implicitly contains somewhat effect of observation error inflation. Is it possible to conduct

bias correction to the RO, like the commonly adopted bias correction for the satellite radiances? Are there systematic features for the bias?

Thank you for your questions and comments. We have revised the manuscript to discuss potential sources of bias, which may arise from the reference analysis, model bias, or the configuration of the DA system. Further optimization of the DA system for these data will be an important focus of our future work.

Bias of different systems (e.g., Met Office, ECMWF, NRL) appears at different levels, suggesting that such biases are model dependent. Even if some bias originates from the observations themselves, its magnitude remains relatively small. We plan to examine this further in our next study.

4. Based on Fig. 8, the authors state that "Notably, there is no clear sign of saturation, as most levels continue to show improvement with increasing numbers of RO profiles. However, the degree of this non-saturation appears to depend on both the variable and vertical level and could be influenced by the specific data assimilation configuration." It seems an overstatement for the 'no clear sign of saturation', especially for the state variables at high levels (e.g., T at 400 hPa, wind speed at 250 hPa). It would be nice to discuss these results. Are the non-saturation errors due to the data assimilation algorithm, or observation type, or other potential reasons?

We agree that the original content included somewhat overstatements. We have made revisions accordingly. For example, we rephrased "Notably, there is no clear sign of saturation" to "Overall, no consistent evidence of saturation was found".

We also mentioned in the revised manuscript that "However, the extent of this non-saturation could be influenced by the specific DA and forecast system". However, since multiple NWP centers also showed positive impacts of increased RO number, it does confirm that RO is a key observation type for such non-saturation impact features.

---

## Author Comment (AC3)

Overview

This paper considers the effect of additional GNSS-RO observations from the ROMEX experiment with the JEDI and GFS systems. Overall it is well-written, provides an interesting summary and deserves to be published. I have a few questions on the presentation of the paper, which need to be addressed. Otherwise it is generally in good shape.

We greatly appreciate your valuable and insightful feedback. In response to the comments, we have reviewed and revised the manuscript accordingly. A point-by-point response is provided below in blue for clarity.

Specific comments

L46: It would be helpful to add Samrat et al (https://rmets.onlinelibrary.wiley.com/doi/10.1002/qj.5002) to the list of publications which show the value of GNSS-RO observations. Since it is a data-denial study it serves to highlight that GNSS-RO is one of the most important observing systems currently available.

Thank you so much for providing this reference. It has been added. It is very informative.

L52: I feel that 2,000 daily profiles from other government missions is an under-estimate. By my counting we have: Metop-B/C (1100 occs), FY-3D/C (1100 occs), Tandem/Terrasar/Grace/PAZ (900 occs, combined), Sentinel-6A (1100 occs). All that comes to around 4,200 profiles per day, unless my calculations are off.

Thank you for your insight! This is also related to the other comments "The Fengyun missions appear to be missing from this list". Fengyun RO data are not available in the U.S. operational data fetch. However, we realize that Fengyun should be counted here. We have made revisions based on the data count given in Anthes et al. (2024) and Marquardt (2024), as well as the statistics in our one-month period.

*Anthes, R. A., C. Marquardt, B. Ruston, and H. Shao, 2024: Radio Occultation Modeling Experiment (ROMEX): Determining the impact of radio occultation observations on numerical weather prediction. Bull. Amer. Meteor. Soc., **105**, 1552–1568. https://doi.org/10.1175/BAMS-D-23-0326.1*

*Marquardt, C., ROMEX data processing. The First ROMEX workshop, EUMETSAT headquarter, Darmstadt, Germany, 17–19 April 2024. https://cdn.eventsforce.net/files/ef-xnn67yq56ylu/website/61/565d7153-abac-414f-92dc-5466867616fc/20240417_13_marquardt_et_al_eumetsat_romex.pdf*

L138: The Fengyun missions appear to be missing from this list.

Thank you for pointing this out. Fengyun has been included in the revised manuscript.

L145: I find this wording confusing "the supplementary profiles are reduced". Perhaps merge this with the previous sentence: "... is 20,000, meaning that ROMEX20K has approximately 12,000 supplementary profiles per day above the BASE experiment."

Thank you for helping improve the readability. Your comment has been incorporated.

L162: I would suggest that the colour scale on Figure 1 is unhelpful, since it is pale in the centre of the range rather than shifting smoothly from light to dark. If possible, please can the authors update this figure?

Thank you for your comments. Figure 1 has been replotted with a different color bar, incorporating also the suggestions of the other two reviewers. The new color bar uses darker colors to indicate higher numbers.

L247: I assume from the large relative errors above 40km that the authors are also using a minimum threshold for the observation error (3 micro-radians is typical). Please can you state what is used.

Thank you very much for pointing out this. Yes, we are using a 3 micro-radians floor. We have missed the description and added in the revised manuscript.

L253: The sentence beginning "All experiments assimilated" is unnecessary, since it is explained in more detail in the following sentence. Please can you remove / reword this sentence.

Thank you for the comments. The sentence and the following one have been combined/revised.

L258: It would be helpful (here or later) to discuss the experimental limitations. The two issues which seem likely to be the largest are the limited experimental period (only one month) and the limited number of other satellite observations used (no hyperspectral IR, geostationary radiances, atmospheric motion vectors, etc.). It would be good to mention these here and in the summary, as well as any other issues of which the authors are aware.

Thank you for your comment. We have added a discussion of these two limitations, along with other relevant limitations, in the final section. We have also state that atmospheric motion vectors are assimilated in our experiments.

L290: Since the acronym MAE is widely used to refer to mean absolute error (similar to the RMSE), the use of MAER could cause some confusion. Perhaps mean absolute bias reduction (MABR) would be a preferable name.

We agree. MABR is used in the revision.

L302: Since Figure 2 goes up to 55 km, I'm surprised that Figure 3 stops at 40 km. Does this imply that the observations are only assimilated to this level (which would need stating if true)? Please could the authors amend the figure, or clarify the assimilation limits?

We do assimilate RO observations from all missions up to 55 km impact height. To clarify, two sentences are added in the figure caption and the content.

"All RO profiles are assimilated from the surface up to 55 km using the same configuration, i.e., the same observation error specification and QC."

In the revised manuscript, we have replotted Fig. 3 up to 55 km and changed the plotted quantity from OMB/O to OMB/B, as OMB/B is more commonly used in RO space verification (e.g., Lonitz 2025; Bowler and Lewis 2025). While the plots appear slightly different from the previous version, the conclusions remain unchanged.

*Bowler, N. E., and O. Lewis: Understanding the impact of additional observations in the Met Office system. The Second ROMEX workshop, EUMETSAT headquarter, Darmstadt, Germany, 25–27 February 2025. https://cdn.eventsforce.net/files/ef-xnn67yq56ylu/website/66/25c71745-2c8d-488b-be58-02f274ecd1c0/7_20250225_neillbowler_metoffice_romex.pdf*

*Lonitz, K., Updates on running ROMEX experiments at ECMWF. The Second ROMEX workshop, EUMETSAT headquarter, Darmstadt, Germany, 25–27 February 2025. https://cdn.eventsforce.net/files/ef-xnn67yq56ylu/website/66/9326ec50-ce3e-47b8-b714-24bafb99d8d8/6_new_20250225_katrinlonitz_ecmwf_romex.pdf*

L379: It is confusing that the figure caption refers to (a) and (c) before (b). Perhaps the individual plots should be reordered so that wind speed appears as Figure 8(b) so that they are in order.

Yes, it makes sense to reorder. Please see the new figure.

L612: Whilst Figure 16 demonstrates a degradation in the standard deviation of forecast error above 50 hPa, the authors speak about sources of the biases. In fact, the changes noted in the presentations from ECMWF and the Met Office largely focused on changes in the forecast bias, rather than the random component of the forecast error. Therefore, it would be helpful to show plots illustrating the change in forecast bias. Additionally, those presentations largely discussed changes in the geopotential height bias in the troposphere, whereas this appears to be a degradation in the stratosphere. It would be helpful for the authors to discuss this difference.

[Figure]

Figure 16: (a) Bias of geopotential height forecast (shaded; unit: geopotential meter or gpm) for BASE verified against ECMWF analysis as a function of forecast lead time for region of NHX (20oN–80oN), and the mean absolute bias (MAB) difference between (b) ROMEX20K and BASE, and (c) ROMEX and BASE. A negative MAB (blue) reflects a beneficial bias reduction relative to BASE, while a positive value (red) indicates a detrimental increase.

Thank you for this very helpful comment. We have replotted Figure 16 and revised the related discussion (see details in the section Summary and Discussion). We have also added remarks on the differences from the results of ECMWF and the Met Office, along with further discussion.

---

## Author Response (AR2)

**Response to the "Remarks from the preceding review file validation"**

1) Your "Short summary" system section includes abbreviations. Please adapt your short summary by avoiding abbreviations to make it better understandable for non-experts and please pay attention to use only 500 characters including spaces.

Please see the revised as below.

This study investigates the impact of increased Global Navigation Satellite System radio occultation (GNSS-RO) profiles on numerical weather prediction (NWP) as part of the Radio Occultation Modeling EXperiment (ROMEX) effort. Leveraging the Joint Effort for Data assimilation Integration (JEDI), the next-generation data assimilation system adopted by NOAA, NASA, NRL and others, this impact study provides critical insights into the necessity of enhanced GNSS-RO observations for NWP applications.

2) Please provide at least the city and country for the authors' affiliation 1.

Done. Please see the revised manuscript; the tracked version has been uploaded as a supplement.

3) Please ensure that the colour schemes used in your maps and charts allow readers with colour vision deficiencies to correctly interpret your findings. Please check your figures using the Coblis – Color Blindness Simulator (https://www.color-blindness.com/coblis-color-blindness-simulator/) and revise the colour schemes accordingly. --> Figs. 4, 5, 6

Thank you for the guidance. Figs. 4-6 have been replotted and carefully checked with <a href="https://www.color-blindness.com/coblis-color-blindness-simulator/">https://www.color-blindness.com/coblis-color-blindness-simulator/</a>

**In addition:**

We have revised the format of the references.